# Effect of High Glucose on Embryological Development of Zebrafish, *Brachyodanio*, *Rerio* through Wnt Pathway

**DOI:** 10.3390/ijms25179443

**Published:** 2024-08-30

**Authors:** Ebony Thompson, Justin Hensley, Renfang Song Taylor

**Affiliations:** 1Division of Natural Sciences, College of Natural and Applied Sciences, Southwest Baptist University, 1600 University Road, Bolivar, MO 65613, USA; ethompson@sbuniv.edu (E.T.); jhensley@sbuniv.edu (J.H.); 2Biology Department, School of Physical Sciences and Engineering, Anderson University, 1100 5th Street, Anderson, SC 29621, USA

**Keywords:** high glucose, embryological development, zebrafish, RNA-seq, metabolic pathway

## Abstract

Gestational diabetes mellitus (GDM) is a worldwide pregnancy complication. Gestational diabetes can significantly impact fetus development. However, the effects of high glucose on embryological development post-fertilization are yet to be researched. *Danio rerio* embryos are a great model for studying embryonic development. In this study, the effects on embryological (morphological and genetic) development were examined in the presence of a high-glucose environment that mimics the developing fetus in pregnant women with GDM. Fertilized zebrafish embryos were treated with normal media and high glucose for 5 days from 3 h post-fertilization (hpf) to 96 hpf, respectively, as control and experimental groups. Morphological changes are recorded with microscope images. Hatch rate and heart rate are compared between groups at set time points. RNA-Seq is performed to examine the gene changes in the experimental group. Glucose delayed the zebrafish embryo development by slowing the hatch rate by about 24 h. The brain, heart, and tail started showing smaller morphology in the glucose group compared to the control group at 24 hpf. Heart rate was faster in the glucose group compared to the control group on days 2 and 3 with a statistically significant difference. Among the zebrafish whole genome, the significantly changed genes were 556 upregulated genes and 1118 downregulated genes, respectively, in the high-glucose group. The metabolic and Wnt pathways are altered under high-glucose conditions. These conditions contribute to significant physiological differences that may provide insight into the functionality of post-embryological development.

## 1. Introduction

Maternal glucose metabolism plays a crucial role in facilitating intrauterine growth. Gestational diabetes mellitus (GDM), a form of diabetes that develops during pregnancy, is characterized by elevated blood sugar levels. It affects approximately 7% of all pregnancies and poses increased risks for both the mother and the fetus. GDM is thought to arise from a combination of insulin resistance and impaired insulin secretion, resulting in hyperglycemia and metabolic abnormalities. The risk factors associated with GDM include obesity, advanced maternal age, a family history of diabetes, and certain ethnicities. GDM can lead to adverse outcomes, including preeclampsia, preterm birth, fetal macrosomia, and neonatal hypoglycemia, affecting the health of both the mother and the baby. GDM has both short-term and long-term effects on maternal health. Globally, GDM affects approximately 9% of pregnancies, increasing the risk of complications during childbirth, cardiovascular disease, and the development of type 2 diabetes later in life [1,2,3,4,5].

The zebrafish (*Danio rerio*) is an important model organism extensively used for studying metabolic mechanism developmental biology [6,7] and genetics [8,9,10] due to its small size, optical transparency, easy treatment, and low-cost caring of the embryo. Zebrafish embryonic development and post-embryonic development are well established for further study of genetic changes in the form of morphological changes [11,12]. Glucose plays a crucial role in the development of zebrafish embryos as it serves as a major energy source for cellular processes. Glucose is metabolized through several pathways in zebrafish embryos, including glycolysis, the pentose phosphate pathway, and the tricarboxylic acid (TCA) cycle. A study on the metabolic profiling of zebrafish embryo development showed that glucose played a vital role during the early stage of zebrafish development [13,14,15,16,17].

Given the critical role of glucose in these metabolic pathways, alterations in glucose availability or metabolism can have significant impacts on zebrafish embryo development. Disruptions in glycolysis or the TCA cycle can lead to decreased ATP production, which can negatively impact cellular processes such as cell division and differentiation. Additionally, alterations in the pentose phosphate pathway can impact NADPH availability, which can impact biosynthesis and oxidative stress defense mechanisms [16,17].

The Wnt signaling pathway is an important regulator of embryonic development in many organisms, including zebrafish [18,19,20,21,22]. This pathway is involved in a wide range of processes, such as cell fate specification, cell proliferation, differentiation, and tissue patterning [23]. In zebrafish, the Wnt signaling pathway is particularly important in early embryonic development, where it plays a key role in the formation and patterning of the dorsal–ventral (DV) axis. Specifically, the Wnt pathway is involved in the specification of the organizer tissue, which is responsible for the induction of the embryonic axes and subsequent tissue patterning. During early embryonic development, the Wnt pathway is activated in the dorsal region of the embryo, where it induces the expression of organizer-specific genes, such as chordin and noggin. These organizer factors, in turn, inhibit the activity of bone morphogenetic proteins (BMPs) that are involved in ventral patterning, leading to the establishment of the DV axis. Furthermore, the Wnt pathway is also involved in cell proliferation and differentiation during embryonic development. For example, in the developing nervous system, Wnt signaling is required for neural progenitor proliferation, differentiation, and axon guidance [21]. The Wnt signaling pathway plays a critical role in zebrafish embryonic development, regulating cell fate specification, tissue patterning, and differentiation. Alterations in this pathway can have significant impacts on embryonic development and can lead to various developmental abnormalities [16,17,18,19].

With the development of sequencing technologies, genetic studies have been dramatically increased to discover the role of key genes and pathway networks during zebrafish development [8,9,10]. Epigenetic studies reveal other key factors in early development including the epigenetic changes in maternal genes [24,25].

The mechanism of glucose and how it affects zebrafish development still need more research to be understood. In this study, we identified genes and pathways regulated under high-glucose treatment using whole-genome expression analysis of RNA in zebrafish embryos. We used gene ontology analysis to identify functional groups of differentially expressed genes. Bioinformatic analyses of our data confirm the impact of high-glucose conditions that mimic GDM on embryonic development.

## 2. Results

### 2.1. Effect of Glucose on Zebrafish Embryo Development

Glucose exposure significantly impacted the developmental timeline of zebrafish embryos. Notably, the hatch rate was delayed by approximately 24 h in the glucose-exposed group compared to the control group, indicating a retardation in embryonic development associated with glucose exposure (Figure 1A).

Evaluation of heart rate revealed significant alterations induced by glucose exposure. On both day 2 and day 3 post-fertilization, embryos in the glucose group exhibited a faster heart rate compared to those in the control group. This difference was statistically significant, highlighting the potential influence of glucose on cardiovascular development in zebrafish embryos (Figure 1B).

Analysis of morphological parameters revealed distinct differences between the glucose-exposed and control groups. At 24 h post-fertilization (hpf), embryos in the glucose group exhibited smaller brain, heart, and tail morphology compared to those in the control group. This suggests that glucose exposure may interfere with the normal growth and development of these vital organs during early embryogenesis (Figure 1C).

### 2.2. RNA-Seq Analysis Displayed Significant Gene Dysregulation in the High-Glucose-Treated Embryos

In this study, we investigated the global transcriptome changes in high-glucose-treated zebrafish embryos using RNA seq. Biological triplicates were performed for the RNA seq. Principal component analysis shows low variability among triplicates (Figure 2A). With about 25 K transcripts, data analysis identified 1674 (6.5%) differentially expressed genes (DEGs) with 556 upregulated and 1118 downregulated, respectively (fold change > 2 and *p* < 0.05) (Figure 2B). The Venn Diagram shows the overlap genes and unique genes in these two groups’ data (Figure 2C). Gene expression differs between the control group and the high-glucose treatment group shown in Figure 2D’s heatmap.

To determine changes in expression levels of individual genes, we identified the top 10 most significant up- and downregulated genes listed in Table 1 and Table 2. The major GO biological processes and molecular functions enriched from the top 10 upregulated genes included the extracellular region, extracellular space, DNA binding transcription factor activity, transcription regulator activity, protein dimerization activity, and NAD binding. The top 10 downregulated genes included enriched processes and molecular functions including the oxidation–reduction process, cofactor biosynthetic process, cofactor metabolic process, oxidoreductase activity, acting on the CH-NH2 group of donors, cofactor binding, acting on peroxide as acceptor, antioxidant activity, and oxygen as an acceptor.

### 2.3. Functional Analysis of DEGs Using Gene Ontology (GO)Analysis

In order to identify the functional associations of the DEGs, Gene Ontology (GO) enrichment analysis was performed by the cluster Profiler R package, in which gene length bias was corrected. GO terms with corrected *p*-values less than 0.05 were considered significantly enriched by differentially expressed genes. Among 410 categories of GO analysis results, 6 categories were significant. Two molecular functions (MFs) and four cellular components (CCs) were identified from the upregulated DEGs. Among 683 categories of GO analysis results, 9 categories were significant. Three biological processes (BPs) and six molecular functions (MFs) were identified from downregulated DEGs. Figure 3A shows the GO analysis for both up- and downregulated DEGs. Both sets of genes are involved in molecular function. Oxidoreductase activity is the main molecular function change in downregulated genes. For the biological process, it was mainly involved in reduction, oxidation, and cofactor-binding activities. In the upregulated genes, the main molecular function is on a transcription and protein change level. The cellular component mainly focuses on extracellular components.

GO terms show significantly enriched Des based on molecular functions. The scatter plot shows the top 30 enriched KEGG pathways in the control and high-glucose groups.

### 2.4. KEGG Pathway Analysis

We analyzed the biological pathways using the Kyoto Encyclopedia of Genes and Genomes (KEGG). KEGG is a database resource for understanding high-level functions and utilities of the biological system, such as the cell, the organism, and the ecosystem, from molecular-level information, especially large-scale molecular datasets generated by genome sequencing and other high-throughput experimental technologies (http://www.genome.jp/kegg/, accessed on 21 April 2022). In total, 725 out of 1118 downregulated DEGs were found in KEGG, and 137 pathways were involved in at least 1 gene. Significant enrichment was found for eight pathways in KEGG: five metabolism pathways, one p53 pathway, one PPAR pathway, and one ABC transporter pathway. In total, 236 genes out of 556 were upregulated DEGs in KEGG, where 96 pathways involved at least one DEG. Significant KEGG enrichment was found for the Wnt and Notch signaling pathways. These pathways are highly consistent with the development of zebrafish embryos and the metabolism system during development. Figure 3B shows the enriched pathways in upregulated and downregulated DEGs.

In total, 97 out of 161 genes were downregulated in the oxidative phosphorylation pathway (Figure 4); 35 out of 60 genes were downregulated in the glycolysis/gluconeogenesis pathway (Figure 5A). These data are consistent with previous studies showing that glucose played a vital role in metabolism during zebrafish embryonic development. Frizzed was significantly upregulated while Fra1 was significantly downregulated in the Wnt/β-catenin pathway (Figure 5B).

## 3. Discussion

In this study, we used 5% D-glucose that mimics the wide fluctuations in glycemia experienced by humans. Following high-glucose treatment, zebrafish embryos exhibited a size change, heartbeat rate changes, changed hatch rates, and survival rate changes. In human cases, the offspring born from mothers with GDM were more obese and conveyed more cardiovascular risks at young ages [20,21].

Zebrafish embryos are a great model to study hyperglycemia-related impact because they can be treated directly with glucose that bypasses any limitations due to the placental–fetal barrier present in mammals [22,23].

The observed delays in hatch rate and alterations in morphological features, particularly in the brain, heart, and tail, suggest that glucose exposure exerts a disruptive effect on zebrafish embryonic development. These findings align with previous studies indicating the susceptibility of developing embryos to metabolic disturbances [14]. The accelerated heart rate observed in the glucose-exposed group further underscores the impact of glucose on cardiovascular function during early development.

The mechanisms underlying these effects warrant further investigation. It is plausible that glucose-mediated alterations in metabolic pathways or signaling cascades may contribute to the observed developmental abnormalities. In the pathway analysis, we found that the pathways related to heart development were changed in the treatment group. Details of the KEGG pathway will be discussed in the following discussion section. Additionally, elucidating the long-term consequences of glucose exposure on zebrafish embryogenesis and subsequent phenotypic outcomes would provide valuable insights into the potential risks associated with elevated glucose levels during critical stages of development.

Our findings highlight the importance of understanding the impact of metabolic factors such as glucose on embryonic development and underscore the potential relevance of zebrafish as a model organism for studying developmental disorders associated with metabolic dysregulation through Wnt signaling.

We performed RNA-seq analysis on high-glucose-treated vs. control zebrafish embryos in order to better understand the molecular-level mechanisms of glucose’s effect on embryonic development through the transcriptome and pathway analysis results. This research will provide clues to further investigate the impact of glucose on GDM’s offspring health condition, diagnosis, and treatments.

Among the KEGG pathways, in all regulated DEGs, 20 pathways were regulated in the high-glucose treatment group and 16 out of those 20 were metabolism pathways, including the biosynthesis of amino acids, fatty acids, and the metabolism by cytochrome p450. Four signaling pathways are related to development, which are the Notch signaling pathway, Wnt signaling pathway, FoxO pathway PPAR, and p53 signaling pathway. There was one ABC transporter pathway. All these pathways have crucial functions during embryo development.

Our investigation into the effects of high glucose levels on zebrafish embryo development revealed significant alterations in key metabolic pathways, with the Wnt signaling pathway shedding light on the intricate relationship between glucose metabolism and embryonic development. Notably, our findings indicate pronounced downregulation of genes involved in glycolysis/gluconeogenesis and oxidative phosphorylation pathways in glucose-exposed embryos compared to controls.

Our findings are consistent with previous studies highlighting the pivotal role of glucose in modulating metabolic pathways during zebrafish embryonic development. High glucose levels have been shown to exert multifaceted effects on cellular metabolism, influencing various metabolic processes essential for embryogenesis. The observed downregulation of genes involved in glycolysis/gluconeogenesis and oxidative phosphorylation pathways corroborates the notion that glucose serves as a central regulator of metabolic homeostasis during critical stages of development.

There are three pathways that regulate heart development and these are significantly regulated in our data. The vascular smooth muscle contraction, cardiac muscle contraction, adrenergic signaling in cardiomyocytes, and regulation of actin cytoskeleton pathways play important roles for normal heart function. These changes indicate that high glucose impacts heart function by changing the molecule level in these.

Wnt signaling plays a crucial role in zebrafish embryonic development, regulating different processes such as axis formation, neurogenesis, neural crest, and organogenesis [23,24]. Wnt signaling regulates the skin and scale development in zebrafish along with the Hedgehog pathway [25]. The Wnt signaling pathway is initiated by the binding of Wnt ligands to Frizzled receptors, leading to the activation of intracellular signaling cascades. One well-studied component of this pathway is β-catenin, which acts as a key transcriptional co-activator. Upon activation, β-catenin translocates to the nucleus, where it interacts with TCF/LEF transcription factors to regulate the expression of target genes involved in embryonic patterning and tissue differentiation. Wnt1, 3, 7, and 10 are the key molecules that stimulate neural progenitor cells. Wnt8 and lef1 are the molecules that are involved in posterior hypothalamic neural-specific development [25].

In our study, the Wnt signaling, Notch, and Hedgehog pathways are significantly changed in the KEGG enrichment pathways. Figure 5B shows that the Frizzled receptor and Fra1 are significantly regulated by high-glucose treatment compared to the control group. The treatment group had sped-up heart rates and abnormal body shape which matches well with the function of the Wnt signaling pathway during zebrafish embryonic development. These findings have the potential to identify early diagnostic biomarkers at the molecular level, which can be utilized in clinical settings. By enabling earlier and more accurate detection, these biomarkers may improve patient outcomes, allowing for timely intervention and personalized treatment strategies tailored to individual needs.

### 3.1. Limitations in This Study

In this study, we compared gene expression changes between the control and glucose-treated groups without accounting for osmolarity differences in the treatment groups. Future research should include the control of osmolarity to ensure more accurate and reliable results.

### 3.2. Implications and Future Directions

The relationship between maternal gene modification under high glucose levels and fetal development gene changes is another way to explore the impact of high glucose levels in mother to embryo development [26,27]. Further investigations are warranted to delineate the precise molecular mechanisms underlying these pathways in both maternal and offspring effects on embryonic development.

### 3.3. Conclusions

In conclusion, our study provides compelling evidence of the profound influence exerted by elevated glucose levels on the Wnt signaling pathway during zebrafish embryo development, as revealed by RNA sequencing (RNA-seq) analysis. The intricate interplay we observed between glucose levels and Wnt signaling underscores the complexity of molecular mechanisms governing embryonic development. This dysregulation not only sheds light on the immediate effects on early developmental processes but also raises intriguing questions about the long-term implications for offspring during subsequent developmental stages. Our findings contribute to a deeper understanding of the molecular pathways involved in glucose-mediated developmental changes, with potential implications for developmental biology and human health. Further investigations into the precise mechanisms by which high glucose levels disrupt Wnt signaling, as elucidated by RNA-seq, will be crucial for unraveling the broader impact on embryonic development and for identifying potential therapeutic interventions to mitigate adverse effects.

## 4. Materials and Methods

### 4.1. Materials

D-glucose (Modernist Pantry 5% *w*/*v*) was used in this study. Embryonic media (E3) was made in our lab using the standard protocol. The experiments design is shown in Figure 6.

### 4.2. Animal Husbandry

Adult wild-type zebrafish were obtained from breeding facilities at Carolina Biological Supply Co. (Whitsett, NC., USA). Fish maintenance, breeding conditions, and egg production were carried out according to the internationally accepted standard protocol [28]. Embryos were obtained from Carolina Biological Supply, USA. Adults were raised in embryonic media (E3) under standard conditions at 28.5 °C with a 14 h light/10 h dark cycle, and the embryonic stages are according to standard procedures [6].

#### 4.2.1. Preparation of D-Glucose Application

Embryos were separated into 5 Petri dishes with each Petri dish containing 20 embryos. At the time of grouping completion, the embryos were ~3 h post-fertilization (hpf). Once all of the 100 embryos had been separated into 5 groups, the collection water was removed and replaced with embryonic media (E3). Embryos were kept in an incubator at 28 °C to wait until they were 5 hpf. 

#### 4.2.2. Addition of D-Glucose and Collection of Embryos

Our three different D-glucose concentrations (2%, 5%, and 10%) were tested. We chose a 5% glucose concentration according to our preliminary data to conduct this study. At this stage, embryos are 5 hpf. Embryos were incubated at 28 °C until they were 32 hpf. Tricaine was used and placed into each of the 5 Petri dishes in order to euthanize each embryo. After a few minutes, each dish was placed under a Leica dissecting scope in order to measure the mortality rate in each group and score for any occurring changes to the embryo. Once the mortality rate was noted, the next step was to remove the chorion of the surviving embryo with surgical-grade forceps. Treatment and control groups underwent the same process. The surviving embryos were placed into individual Eppendorf tubes and three drops of 4% PFA were added in order to conserve the embryos’ structure.

#### 4.2.3. Hatch Rate Calculation

In the treatment and control groups, hatched larvae were counted, and the hatching rate (%) was calculated by hatched numbers/total exposed numbers × 100 [29].

### 4.3. Total RNA Extraction

The control and glucose-treated zebrafish embryos were collected for total RNA extraction. The experiment design includes 3 control samples and 3 high-glucose-treated samples. A total of 6 RNA samples were extracted using a Zymo RNA extraction kit (Irvine, CA, USA) according to the manufacturer’s protocol. The concentration of the RNA samples was measured by a Nano drop 2000 (DeNovix Inc., Wilmington, DE, USA).

### 4.4. RNA Seq

RNA seq was performed by experts at Novogene Corporation Inc. (West Coast: Sacramento, CA, USA). The sample quality control, library preparation, clustering, and sequencing are in the supplemental methods.

RNA seq data analysis was performed by the Novogene Corporation Inc. (West Coast: Sacramento, CA, USA).

Briefly, raw data (raw reads) of Fastq format were first processed through in-house per scripts. In this step, clean data (clean reads) were obtained by removing reads containing adapter sequences, reads containing ploy-N, and low-quality reads from raw data. At the same time, the Q20, Q30, and GC contents of the clean data were calculated. All the downstream analyses were based on clean data of high quality.

### 4.5. Reads Mapping to the Reference Genome

Reference genome and gene model annotation files were downloaded from the genome website directly. The index of the reference genome was built using Hisat2 v2.0.5 and paired-end clean reads were aligned to the reference genome using Hisat2 v2.0.5. We selected Hisat2 as the mapping tool because Hisat2 can generate a database of splice junctions based on the gene model annotation file and thus a better mapping result than other non-splice mapping tools.

### 4.6. Novel Transcript Prediction

The mapped reads of each sample were assembled by StringTie (v1.3.3b) in a reference-based approach [30]. StringTie uses a novel network flow algorithm as well as an optional de novo assembly step to assemble and quantitate full-length transcripts representing multiple splice variants for each gene locus.

### 4.7. Quantification of Gene Expression Level

FeatureCounts v1.5.0-p3 was used to count the read numbers mapped to each gene. Subsequently, the FPKM of each gene was calculated based on the length of the gene and the read count mapped to this gene. FPKM, the expected number of Fragments Per Kilobase of transcript sequence per Million base pairs sequenced, considers the effect of sequencing depth and gene length for the read count at the same time and is currently the most commonly used method for estimating gene expression levels.

### 4.8. Differential Expression Analysis

For DESeq2 with biological replicates, differential expression analysis of two conditions/groups (two biological replicates per condition) was performed using the DESeq2R package (1.20.0). DESeq2 provides statistical routines for determining differential expression in digital gene expression data using a model based on the negative binomial distribution. The resulting *p*-values were adjusted using Benjamini and Hochberg’s approach for controlling the false discovery rate. Genes with an adjusted *p*-value ≤ 0.05 found by DESeq2 were assigned as differentially expressed. Prior to differential gene expression analysis, for each sequenced library, the read counts were adjusted by the edge R program package through one scaling normalized factor. Differential expression analysis of two conditions was performed using the edge R package (3.22.5). The *p*-values were adjusted using the Benjamini–Hochberg method. A corrected *p*-value of 0.05 and an absolute fold change of 2 were set as the thresholds for significantly differential expression.

### 4.9. GO and KEGG Enrichment Analysis of Differentially Expressed Genes

Gene Ontology (GO) enrichment analysis of differentially expressed genes was implemented by the cluster Profiler R package, in which gene length bias was corrected. GO terms with corrected *p*-values less than 0.05 were considered significantly enriched by differentially expressed genes. We used the cluster Profiler R package to test the statistical enrichment of differential expression genes in KEGG pathways.

### 4.10. Gene Set Enrichment Analysis

Gene set enrichment analysis (GSEA) is a computational approach to determine if a predefined gene set can show a significant consistent difference between two biological states. The genes were ranked according to the degree of differential expression in the two samples, and then the predefined gene sets were tested to see if they were enriched at the top or bottom of the list. Gene set enrichment analysis can include subtle expression changes. We used the local version of the GSEA tool (http://www.broadinstitute.org/gsea/index.jsp, accessed on 21 April 2022) and GO and KEGG datasets were used for GSEA independently.

## Figures and Tables

**Figure 1 ijms-25-09443-f001:**
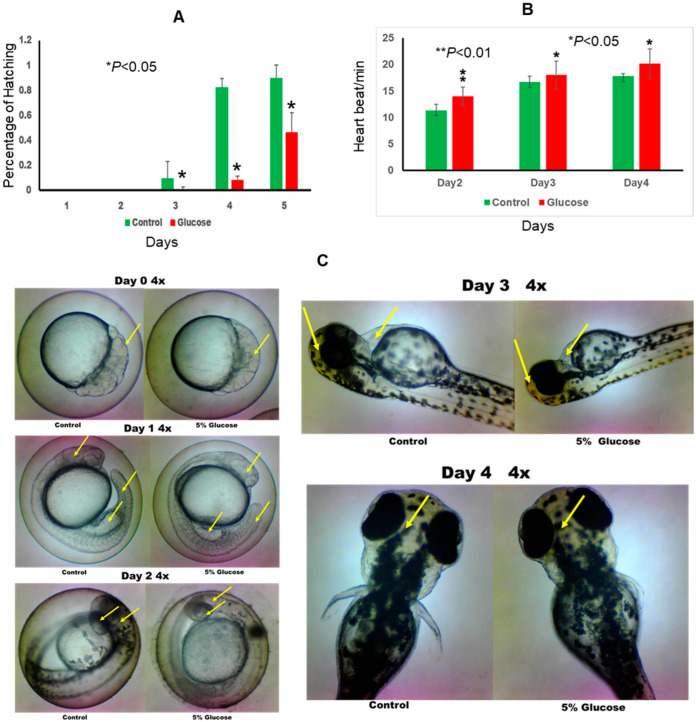
Effect of glucose on zebrafish embryos. (**A**) The hatch rate of control and glucose groups. (**B**) The heart rate of control and glucose groups. (**C**) The morphology changes of control and glucose groups. Arrows indicate the changes and show a comparison of the two groups at different times of treatment.

**Figure 2 ijms-25-09443-f002:**
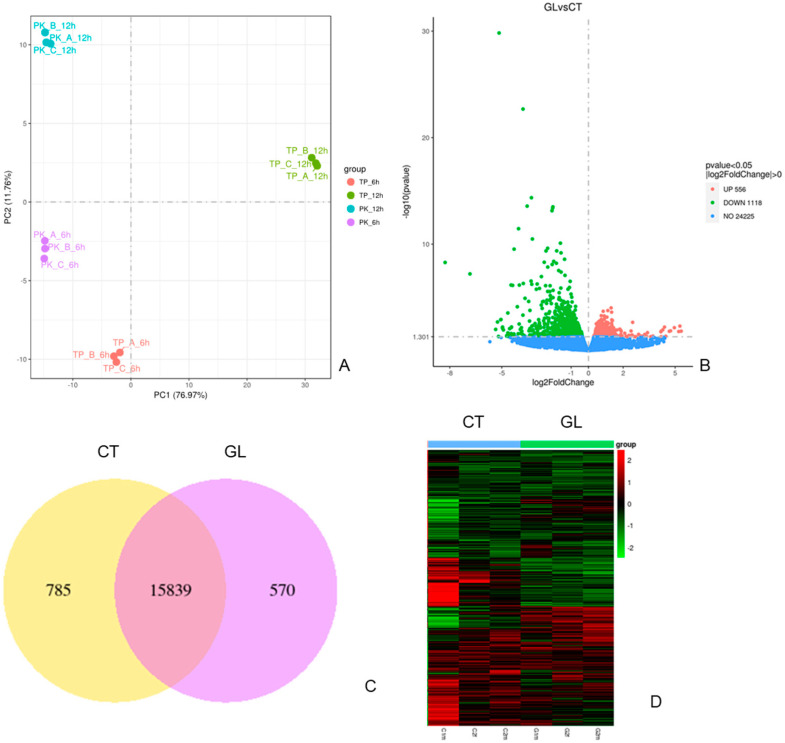
RNA seq data analysis with control and glucose treatment groups. (**A**) PCA analysis within treatment groups. (**B**) Volcano plots for gene changes. Upregulated genes are shown in red and downregulated genes are shown in green. (**C**) Gene differential expression in control and glucose groups. (**D**) Heatmap showing gene expression intensity. Red shows higher expression and green shows lower expression in RNA-seq levels.

**Figure 3 ijms-25-09443-f003:**
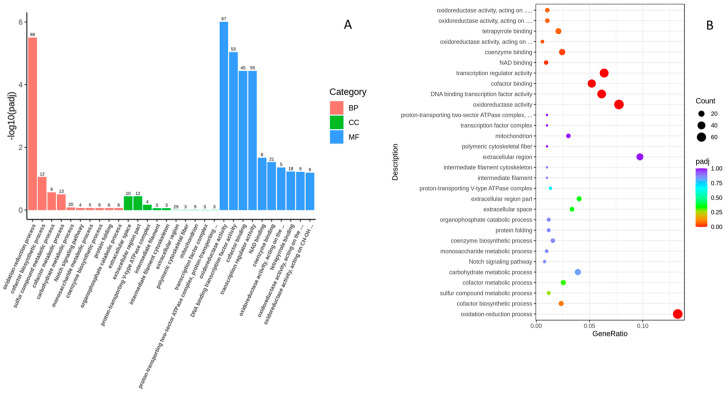
(A) GO enrichment pathway analysis of up- and downregulated DEGs. (**B**) The scatter plot of top 30 enriched KEGG pathways.

**Figure 4 ijms-25-09443-f004:**
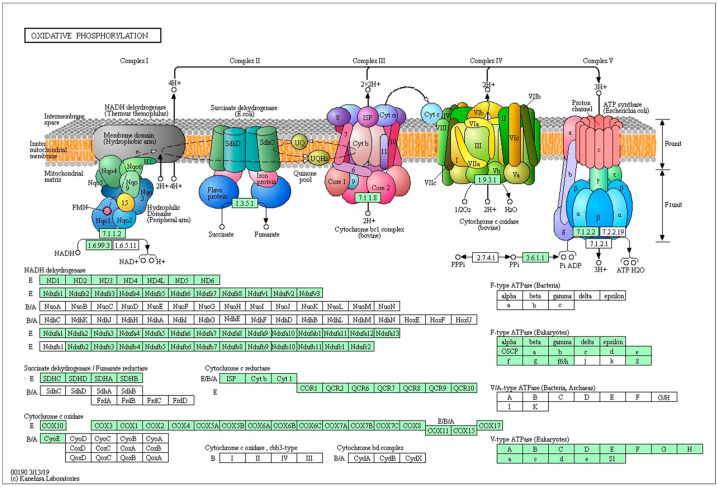
The pathway of oxidative phosphorylation (the downregulated genes are shown in green).

**Figure 5 ijms-25-09443-f005:**
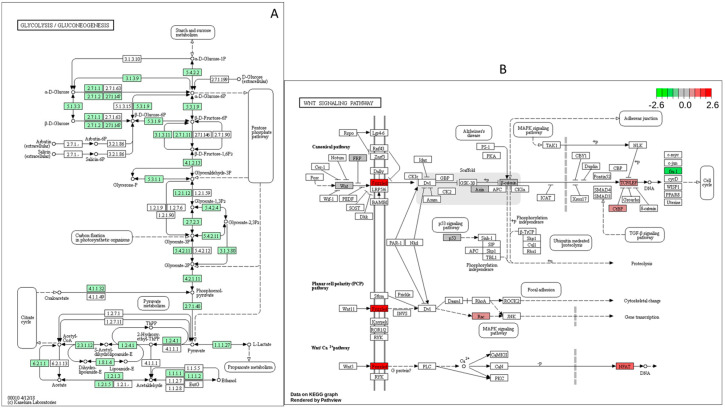
KEGG pathways from RNA seq data analysis (the downregulated genes are shown in green and upregulated genes are shown in red). (**A**) The glycolysis/gluconeogenesis pathway. (**B**) The Wnt signaling pathway.

**Figure 6 ijms-25-09443-f006:**
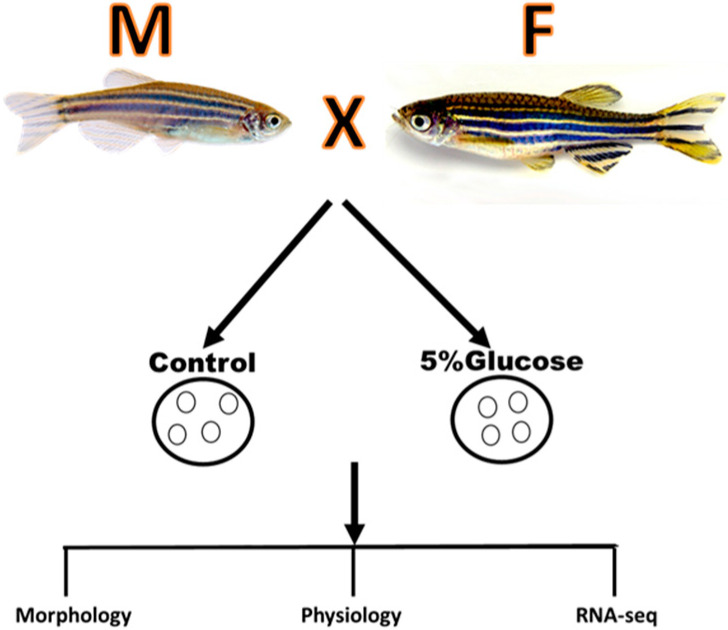
The schematic of experimental design on the effect of glucose on zebrafish embryos.

**Table 1 ijms-25-09443-t001:** Top 10 upregulated genes in glucose-treated zebrafish embryos in RNA Seq.

Gene	Biological Process	Molecular Function
*Kcnj1a.2*	Transport; ion transport	Enable ATP-activated inward rectifier potassium channel activity
*Eomesa*	Enables DNA binding activity and transcription factor activity in blastoderm, blastodisc, brain, eye, and immune system	DNA-binding and DNA-binding transcription factor activity
*zgc:77439*	Enables N,N-dimethylaniline monooxygenase activity, NADP binding activity and flavin adenine dinucleotide binding activity	NADP binding activity in response to xenobiotic stimulus in endoplasmic reticulum
*foxg1d*	Transcription regulation, DNA binding activity	Enables sequence specific double-stranded DNA binding activity in nucleus
*zgc:15840*	Transport; urea transmembrane transporter activity	Enables serine-type endopeptidase activity and urea transmembrane transporter activity
*tbr1b*	Commitment of neuronal cells to specific neuron type in forebrain, neuron projection development, transcription regulation	DNA-binding transcription factor activity
*pcdh1g32*	Calcium ion binding activity and cell adhesion	Homophilic cell adhesion via plasma membrane adhesion molecules in the brain
*tert*	Telomerase activity; erythrocyte maturation in brain, eye, hear, liver, pleuroperitoneal region, intestinal epithelial structure maintenance	Telomere maintenance via telomerase
*plp1a*	myelination	Structural constituent of myelin sheath
*TIAM2*	GTPase activity, axonogenesis regulation, signal transduction	enables GTPase activator activity and guanyl-nucleotide exchange factor activity

**Table 2 ijms-25-09443-t002:** Top 10 downregulated genes in glucose-treated zebrafish embryos in RNA Seq.

Gene	Biological Process	Molecular Function
*apoda.2*	lipid metabolic process, response to reactive oxygen species, aging, lipid transport	Enables lipid binding activity
*lct*	Carbohydrate metabolic processes, hydrolase activity	Hydrolyzes O-glycosyl compounds
*tnfrsf11b*	Response to mechanical stimulus	Tumor necrosis factor receptor; head and scale
*opn5*	Absorption of visible light	Enables 11-cis retinal binding activity and all-trans retinal binding activity
*rgra*	G protein-coupled receptor signaling, cellular response to light stimulus and phototransduction	Enables G protein-coupled photoreceptor activity
*zgc:153154*	Small molecule binding in extracellular region	Small molecule binding activity
*cd74b*	Ear development and neuromast development	MHC class II protein binding activity, cytokine receptor activity, macrophage migration inhibitory factor binding activity
*zgc:153911*	T cell receptor signaling, cytokine production regulation	Signaling receptor binding activity in membrane
*pah*	Cellular response to estrogen stimulus in structures incl. Digestive system, epidermis, eye, optic vesicle, and periderm	Phenylalanine 4-monooxygenase activity
*rasef*	GDP and GTP binding in cytosol and perinuclear region of cytoplasm	Enables GDP and GTP binding activity

## Data Availability

We would like to share the data published in this article.

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
