# Peer review of "Effect of High Glucose on Embryological Development of Zebrafish, Brachyodanio, Rerio through Wnt Pathway"

_ijms, 2024, doi:10.3390/ijms25179443_

Round 1

Reviewer 1 Report

Comments and Suggestions for Authors

The article addresses an interesting topic. However, there are several questions that need to be completed.

- In the legend to Figure 1, indicate what test statistics were used and what statistics are presented in graphs A and B (mean-standard error?).

- In methods, indicate how the hatch rate was calculated.

- In methods, add a statistical analysis section.

- Why were the 10 differentially expressed genes not validated with PCR?

- There are many spelling and formatting errors throughout the article, check carefully, e.g. in lines 226, 310, 316

Author Response

Please see the attachment for response to Reviewer 1. Thank you very much!

Reviewer 2 Report

Comments and Suggestions for Authors

I appreciate the opportunity to review the manuscript entitled “Effect of High Glucose on Embryological Development of Zebrafish, Brachyodanio rerio through Wnt Pathway” submitted in journal International Journal of Molecular Sciences.  

The authors estimated the effect of the high glucose on the some morphological caracteristics of the Zebra Fish embryos.

Reviewer Comments:

1.      Please prepare list of abbreviations used in manuscript.

2.      The majority of the references are older than 5 years. Please input the current references.

3.      Please add the paragraphs about impact of the diabetes mellitus and high glucose conditions on the female reproductive system especially pregnancy.

4.      Please add the paragraphs about potential clinical usage of yours results.

Taking into account the pathology reviewed in above mentioned manuscript, my opinion is that this submission meets the criteria to be published in journal International Journal of Molecular Sciences after minor revisions and inclusion of the data I suggested.

Author Response

Please see the attachment. Thank you so much for your comments.

Reviewer 3 Report

Comments and Suggestions for Authors

My comments:

The table 1 and Table 2 have the same title.

The research question should be specific, such as the impact on the development of cardiovascular or central or peripheral nervous system and focus on an unresolved problem and a signaling pathway or the molecules that may be involved.

To support the results related, two to three functional experiments should be performed.

Author Response

Thank you very much for your valuable comments. Please see the attachment for the response. Thank you.

Round 2

Reviewer 3 Report

Comments and Suggestions for Authors

It makes sense for the response to the reviewer’s comments.